# Adaptive Conformal Prediction for Quantum Machine Learning

**Douglas Spencer**                                          *douglas.spencer@reuben.ox.ac.uk*
*Mathematical Institute, University of Oxford*

**Samual Nicholls**                                          *samual.nicholls@ccc.ox.ac.uk*
*Mathematical Institute, University of Oxford*

**Michele Caprio**                                          *michele.caprio@manchester.ac.uk*
*Department of Computer Science, The University of Manchester*

**Reviewed on OpenReview:** *https://openreview.net/forum?id=ShkPB9OeEW*

## Abstract

Quantum machine learning seeks to leverage quantum computers to improve upon classical machine learning algorithms. Currently, robust uncertainty quantification methods remain underdeveloped in the quantum domain, despite the critical need for reliable and trustworthy predictions. Recent work has introduced quantum conformal prediction, a framework that produces prediction sets that are guaranteed to contain the true outcome with a user-specified probability. In this work, we formalise how the time-varying noise inherent in quantum processors can undermine conformal guarantees, even when calibration and test data are exchangeable. To address this challenge, we draw on Adaptive Conformal Inference, a method which maintains validity over time via repeated recalibration. We introduce Adaptive Quantum Conformal Prediction (AQCP), an algorithm which provides asymptotic average coverage guarantees under arbitrary hardware noise conditions. Empirical studies on an IBM quantum processor demonstrate that AQCP achieves the target coverage level and exhibits greater stability than quantum conformal prediction.

## 1 Introduction

Quantum machine learning (QML) aims to integrate quantum algorithms into broader machine learning pipelines, seeking performance advantages over classical methods on specialised tasks (Biamonte et al., 2017). QML techniques have already proven effective across supervised, unsupervised, and reinforcement learning settings (Havlíček et al., 2019; Otterbach et al., 2017; Jerbi et al., 2021).

Classical machine learning models are increasingly deployed in domains with significant societal implications, for example, in justice, healthcare, transportation, and defence (Završnik, 2020; Shailaja et al., 2018; Lindemann et al., 2023; Svenmarck et al., 2018). In such high-stakes settings, erroneous decisions can carry severe consequences, making rigorous uncertainty quantification essential to model evaluation and deployment.

For QML models to become trustworthy tools, they must also incorporate reliable uncertainty quantification. This challenge is heightened by the hardware noise in current noisy intermediate-scale quantum (NISQ) devices, which may vary in both character and severity over time (Proctor et al., 2020).

Conformal prediction stands out as a technique for uncertainty quantification, offering distribution-free, finite-sample guarantees on predictive performance (Angelopoulos & Bates, 2023). Its principal function is to post-process predictive model outputs to produce prediction sets that encompass the true outcome with a user-specified probability. Conformal prediction is appealing within the machine learning community for its ability to function as a wrapper around black-box models (Caprio, 2025; Caprio et al., 2025b).

A commonly used variant, known as split conformal prediction, was recently extended to quantum models in Park & Simeone (2023). Their seminal work introduces the quantum conformal prediction (QCP) framework, which is designed to use measurements from parametrised quantum circuits (PQCs). When tested on quantum hardware, QCP consistently maintained the target coverage level (Park & Simeone, 2023). However, the theoretical guarantees of QCP require a simplifying assumption: that the noise processes within the quantum hardware remain stationary over time.

In this work, we demonstrate that, without this assumption, one can no longer assert that conformity scores are exchangeable, a property that is necessary for standard conformal guarantees. To address this, we extend Adaptive Conformal Inference (Gibbs & Candès, 2021) to the quantum setting, yielding the Adaptive Quantum Conformal Prediction (AQCP) algorithm. This algorithm incorporates online recalibration to explicitly handle non-stationary hardware noise and exhibits greater stability than QCP in experiments using an IBM quantum processor. AQCP is straightforward to implement and provides a reliable representation of uncertainty for QML models under arbitrary hardware noise conditions. Alongside demonstrating AQCP's effectiveness, we provide an experimental analysis of various score functions, comparing the average set size they induce when used with AQCP.

## 1.1 Related Work

Beyond Park & Simeone (2023), which we introduce above, Tasar (2025b) explored an alternative use of conformal prediction for PQCs. In their method, conformal prediction is applied to classical models trained to emulate PQC output distributions using features derived from circuit architecture and gate frequencies. Additionally, Tasar (2025a) investigates conformal prediction in a wide variety of quantum settings, including whether conformal prediction can be used to detect entanglement, the impact of context-conditional exchangeability, and an application to anomaly detection. While these works demonstrate that conformal prediction can be meaningfully applied in quantum contexts, none directly examine how non-stationary model noise affects the exchangeability of conformity scores.

In the classical setting, a growing body of literature has sought to relax the exchangeability requirement between calibration and test data to handle settings such as time series and covariate shift. These methods generally address either specific, known distributional shifts or more general deviations from exchangeability (Tibshirani et al., 2019; Foygel Barber et al., 2023; Gibbs et al., 2025; Gibbs & Candès, 2021). We take inspiration from this literature in our approach.

The design of score functions for quantum conformal prediction closely aligns with that of probabilistic conformal prediction (Wang et al., 2023). Park & Simeone (2023) employ a k-nearest neighbour (k-NN) score function, building upon the approach in Wang et al. (2023), who propose a sampling strategy combined with a 1-nearest neighbour (1-NN) score function. Wang et al. (2023) have also influenced methods for other probabilistic models, including the Conformal-Predict-Then-Optimise (CPO) framework (Patel et al., 2024), which similarly employs a k-NN approach. Sample-based strategies have also appeared in the context of conformal risk control (Zecchin et al., 2023), where samples from the model distribution are used to generate prototypical sequences, to which a Euclidean distance score function is then applied.

## 1.2 Contributions

This paper presents a rigorous framework for reliable uncertainty quantification in quantum machine learning in the presence of realistic, non-stationary hardware noise. Our primary contributions are:

- **Formalisation of time dependence:** We develop a theoretical framework demonstrating how non-stationary noise invalidates the exchangeability of conformity scores, even when calibration and test data are exchangeable.

- **Adaptive algorithm for quantum conformal prediction:** We adapt the Adaptive Conformal Inference method (Gibbs & Candès, 2021) to the quantum machine learning setting. This approach, which we term Adaptive Quantum Conformal Prediction (AQCP), utilises score functions specifically designed to operate on samples from an implicit probability distribution.

- **Comprehensive hardware evaluation:** We conduct a thorough experimental evaluation of AQCP's validity and efficiency on IBM quantum hardware, analyse the performance of various sample-based score functions, and show that AQCP maintains the target coverage level with greater stability than the QCP framework.

## 2 Background

### 2.1 Conformal Prediction

There are two main variants of conformal prediction: full and split. Full conformal prediction is the original variant and is the most data-efficient (Vovk et al., 2005; Shafer & Vovk, 2008; Caprio et al., 2025a). Here we focus on split conformal prediction for its computational efficiency, and we will frequently refer to split conformal prediction as simply conformal prediction.

In split conformal prediction, prediction sets are constructed from a trained model, a calibration dataset $\mathcal{D}_{\text{cal}} = \{(x_i, y_i)\}_{i=1}^n$, and a test feature $x_{n+1}$. For each element of $\mathcal{D}_{\text{cal}}$, a real-valued score is computed, with higher scores assigned when the model's prediction conforms less to the target. Then, for each element $y$ of the target space, a candidate score is computed from $(x_{n+1}, y)$, and inclusion in the prediction set is determined by comparing this score to the calibration scores. Specifically, for a desired miscoverage rate $\alpha \in [0, 1]$ and score function $\hat{S} : \mathcal{X} \times \mathcal{Y} \to \mathbb{R}$, the procedure is as follows:

1. Compute the calibration score $s_i = \hat{S}(x_i, y_i)$ for each calibration point $(x_i, y_i) \in \mathcal{D}_{\text{cal}}$.

2. Set $\lambda$ equal to the $\lceil (n+1)(1-\alpha) \rceil$-th smallest value among $s_1, \ldots, s_n, +\infty$.

3. For a given test input $x_{n+1}$, construct the prediction set:

$$C(x_{n+1}) := \left\{ y \in \mathcal{Y} : \hat{S}(x_{n+1}, y) \leq \lambda \right\}.$$

In what follows, we use uppercase letters (e.g., $X_i, Y_i, S_i$) to denote random variables and distinguish them from their realised values (e.g., $x_i, y_i, s_i$).

Under the weak assumption that the calibration and test points are exchangeable, meaning that their joint distribution is invariant under permutations of the indices (see Section 2.1.1), the following marginal coverage guarantee holds.

**Theorem 1 (Vovk et al. (2005); Lei et al. (2017))** *If $(X_i, Y_i)$, $i = 1, \ldots, n$ are exchangeable, then for a new exchangeable draw $(X_{n+1}, Y_{n+1})$,*

$$\mathbb{P}(Y_{n+1} \in C(X_{n+1})) \geq 1 - \alpha.$$

*Additionally, if the scores $S_1, \ldots, S_n$ have continuous joint distribution, then we have*

$$\mathbb{P}(Y_{n+1} \in C(X_{n+1})) \leq 1 - \alpha + \frac{1}{n+1}.$$

Here, the lower bound arises from Vovk et al. (2005) and the upper bound from Lei et al. (2017).

#### 2.1.1 The Role of Exchangeability in Split Conformal Prediction

Exchangeability is the cornerstone of conformal prediction. This section gives an informal proof of the lower bound of Theorem 1 in the case of almost surely distinct scores, with particular attention to the role of exchangeability.

A finite set of random variables, $Z_1, \ldots, Z_{n+1}$, is said to be exchangeable if their joint distribution is invariant under any permutation of the indices. Formally, for any permutation $\sigma \in \text{Sym}(n+1)$ (the permutation group of order $n + 1$), we require that

$$(Z_1, \ldots, Z_{n+1}) \stackrel{d}{=} (Z_{\sigma(1)}, \ldots, Z_{\sigma(n+1)}),$$

where $\overset{d}{=}$ denotes equality in distribution. This property is weaker than the i.i.d. assumption but implies that the order of the variables carries no statistical information.

In the context of split conformal prediction, we consider a calibration dataset $\{(X_i, Y_i)\}_{i=1}^n$ and a new test point $(X_{n+1}, Y_{n+1})$, where $Y_{n+1}$ is unknown. If these $n+1$ pairs are exchangeable and we apply a fixed score function $\hat{S}(\cdot, \cdot)$ to each, then the resulting conformity scores $S_1, \ldots, S_{n+1}$ are also exchangeable.

This preservation follows directly from Kuchibhotla (2020, Theorem 3). The theorem shows that a transformation $G : (\mathcal{X} \times \mathcal{Y})^{n+1} \to \mathbb{R}^{n+1}$ preserves exchangeability if it satisfies a specific permutation-equivariance condition; for any permutation $\pi_1 \in \mathrm{Sym}(n+1)$, there exists a corresponding permutation $\pi_2 \in \mathrm{Sym}(n+1)$ such that

$$\pi_1 G(w) = G(\pi_2 w), \quad \forall\, w \in (\mathcal{X} \times \mathcal{Y})^{n+1}.$$

In our setting, $G$ corresponds to the map that assigns scores to data points, i.e. $G((X_i, Y_i)_{i=1}^{n+1}) = (\hat{S}(X_i, Y_i))_{i=1}^{n+1}$. Because $\hat{S}$ is applied identically to every calibration and test point, $G$ trivially satisfies the permutation condition, and thus the conformity scores inherit exchangeability from the data.

The key insight is that exchangeability enables a probabilistic argument through ranking. Specifically, because the scores $(S_1, \ldots, S_{n+1})$ are exchangeable, the rank of $S_{n+1}$ is uniformly distributed on $\{1, 2, \ldots, n, n+1\}$. Therefore,

$$\mathbb{P}(\mathrm{rank}(S_{n+1}) \leq \lceil (1-\alpha)(n+1) \rceil) = \frac{\lceil (1-\alpha)(n+1) \rceil}{n+1} \geq 1 - \alpha.$$

Let $\tilde{\lambda}$ denote the $\lceil (1-\alpha)(n+1) \rceil$-th smallest value among $\{S_1, \ldots, S_{n+1}\}$. We can rewrite the event

$$\{\mathrm{rank}(S_{n+1}) \leq \lceil (1-\alpha)(n+1) \rceil\} = \{S_{n+1} \leq \tilde{\lambda}\}.$$

Furthermore, we can remove dependence on the test score by defining $\lambda$ as the $\lceil (1-\alpha)(n+1) \rceil$-th smallest value among $\{S_1, \ldots, S_n, +\infty\}$, and observing that

$$S_{n+1} \leq \tilde{\lambda} \quad \Longleftrightarrow \quad S_{n+1} \leq \lambda.$$

Here, the forward implication follows from $\tilde{\lambda} \leq \lambda$, and the reverse from the fact that $S_{n+1} > \tilde{\lambda}$ implies $\lambda = \tilde{\lambda}$. Hence, for any significance level $\alpha \in [0, 1]$, we obtain

$$\mathbb{P}(S_{n+1} \leq \lambda) \geq 1 - \alpha.$$

As a result, defining $C(X_{n+1})$ as in Section 2.1 gives the lower bound of Theorem 1. Intuitively, this guarantee holds because exchangeability ensures that the test point has no special status amongst the calibration data. For more in-depth treatments, see Vovk et al. (2005); Angelopoulos & Bates (2023).

## 2.2 Quantum Machine Learning

In this work, we employ the classical-data quantum-processing (CQ) paradigm of quantum machine learning, as introduced in Simeone (2022). In this paradigm, classical data are fed into a quantum model, which is trained using a classical optimiser. We use the term *quantum model* to refer to a parametrised quantum circuit (PQC). A PQC applies a unitary transformation $U(\theta)$, dependent on a vector of tunable parameters $\theta$, to a quantum state that encodes a classical input $x$ (Simeone, 2022).

For a working understanding of the CQ paradigm, two key components warrant further explanation: the design of the unitary transformation via the construction of a PQC, and the encoding of the classical input $x$ into the circuit, referred to as quantum data encoding (Rath & Date, 2024; Schuld, 2021). For a more general introduction to the QML landscape, see Chang & Cerezo (2025).

### 2.2.1 PQC Ansätze

In quantum computing, an ansatz defines the structure of a quantum circuit by specifying both the set of gates used and their configuration. This selection is analogous to choosing a model architecture in classical machine

learning, where the design profoundly influences the capability and efficiency of the model (Benedetti et al., 2019). Current research efforts focus on identifying optimal ansätze for various applications, particularly in the context of variational quantum algorithms (Qin, 2023). The hardware-efficient ansatz is a subclass of ansatz designs that mitigates the gate overhead incurred during circuit compilation (Leone et al., 2024). It does so by reducing idle qubits and employing native entangling gates inherent to the hardware. This design is especially well suited to today's NISQ devices, where circuit depth and fidelity are constrained.

The hardware-efficient ansatz is constructed as a sequence of layers, each consisting of local parametrized single-qubit gates applied in parallel to every qubit, followed by a fixed entangling-gate pattern applied between specified qubit pairs. In many implementations, the local gates are chosen to be rotations about the Pauli axes, denoted $R_X(\theta)$, $R_Y(\theta)$, and $R_Z(\theta)$, respectively, where $\theta$ is the rotation angle. The entangling block is typically composed of a fixed pattern of CNOT (CX) or CZ gates that reflect the hardware's connectivity graph. Common entanglement schemes include linear (chain), circular, and all-to-all (full) connectivity (Simeone, 2022). See Figure 1 for circuit diagrams of these three entangling-block configurations, each implemented using CZ gates.

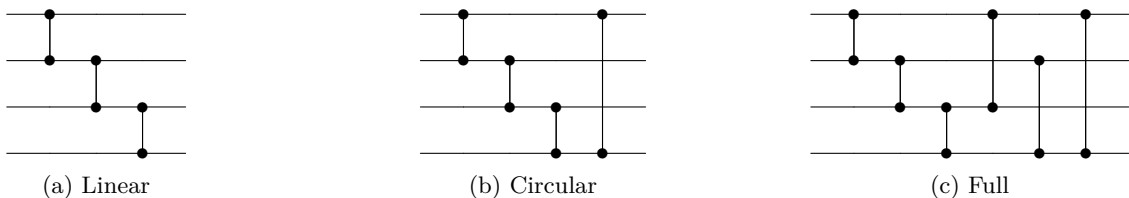

(a) Linear           (b) Circular           (c) Full

Figure 1: Diagrams of three entangling-block configurations within a four-qubit circuit implemented using CZ gates: (a) a linear entangling block, (b) a circular entangling block, and (c) a full entangling block.

### 2.2.2 Angle Encoding

In any supervised quantum machine learning algorithm, the classical data must be encoded into the quantum state prepared by the PQC. This encoding is an essential step in any CQ framework and can be achieved in several ways. Widely used strategies include basis encoding, which maps classical bits directly onto computational-basis states; amplitude encoding, which embeds a normalised feature vector into the probability amplitudes of a quantum state; and angle encoding, in which data values determine the rotation angles of quantum gates (Rath & Date, 2024; Simeone, 2022).

We focus on angle encoding for its relevance to our experimental implementation and that of Park & Simeone (2023). An angle encoder maps a classical feature vector to a set of rotation angles, each of which parametrises a single-qubit rotation gate within the PQC. In the simplest setting, an $n$-dimensional input $(x_1, \ldots, x_n)$ is encoded by applying single-qubit rotations on $n$ qubits, where the rotation angle assigned to the $i^{\text{th}}$ qubit is determined by the corresponding feature $x_i$. These local rotations prepare a product state $|\psi(x)\rangle$ whose representation in the Hilbert space reflects the structure of the input vector. Entanglement, if required, is typically introduced through subsequent entangling layers rather than the encoding itself. This embedding strategy implicitly defines a quantum kernel, dictating the expressivity and feature space of the quantum model (Schuld, 2021).

### 2.3 Density Matrices and Noise Channels

To accurately describe quantum model operations and noise in the next section, we require density-matrix and noise-channel formalism, which we briefly introduce here. For a more comprehensive treatment, see Keyl (2002); Wilde (2013); Fano (1957).

A pure quantum state can be described in two equivalent ways: as a state vector $|\psi\rangle$ in a Hilbert space, or more generally as a density matrix $\rho$. While state vectors provide a convenient representation for pure states, density matrices extend this representation to mixed states, which are probabilistic mixtures of pure

states. Formally, a density matrix is a positive semidefinite, Hermitian operator with unit trace,

$$\rho = \sum_i p_i \left|\psi_i\right\rangle \left\langle\psi_i\right|, \quad \sum_i p_i = 1,$$

where $p_i$ is the probability that the system is prepared in the pure state $\left|\psi_i\right\rangle$. This formalism is essential for modelling realistic quantum systems, as noise and decoherence inevitably lead to mixed states. In the context of QML, density matrices are particularly useful for analysing how data encoding and variational circuits interact with hardware noise.

In current quantum hardware, quantum states are unavoidably affected by noise processes such as decoherence, gate errors, and measurement imperfections. These noise processes can be modelled as quantum channels, mathematically described by completely positive trace-preserving (CPTP) maps acting on density matrices (Nielsen & Chuang, 2010, Section 8.3). A quantum channel $\mathcal{E}$ transforms a state $\rho$ as

$$\rho \mapsto \mathcal{E}(\rho) = \sum_k E_k \rho E_k^\dagger,$$

where $\{E_k\}$ are Kraus operators satisfying $\sum_k E_k^\dagger E_k = I$, and $E_k^\dagger$ denotes the conjugate transpose of $E_k$. This operator-sum representation, known as the Kraus representation, provides a powerful and general framework for capturing the effects of noise on quantum computations.

Several standard noise channels are commonly used to model realistic quantum hardware (Nielsen & Chuang, 2010):

- **Depolarising channel:** With probability $p$, the state is replaced by the maximally mixed state, modelling uniform random errors.

- **Phase flip channel:** Randomly introduces phase flips, capturing loss of coherence without affecting populations.

- **Amplitude damping channel:** Describes energy dissipation processes, such as spontaneous emission from an excited state to the ground state.

## 3 Conformal Prediction Under Non-Stationary Model Noise

Quantum models are also inherently noisy. While it is well established that quantum hardware experiences noise, the extent and nature of non-stationary noise remain active areas of research (He et al., 2024; Proctor et al., 2020). Much of the existing analysis focuses on the gate level, with comparatively less work at the level of full circuits. Notable recent works by Dasgupta & Humble (2020; 2021; 2022) explore this issue in detail.

To mitigate the effects of drift, IBM Quantum systems perform both hourly and daily recalibrations (IBM Quantum, 2025). Furthermore, many current experimental demonstrations rely on recalibrating quantum computers immediately before execution and adjusting them during runtime (He et al., 2024). Sources of non-stationary noise are diverse, including temperature fluctuations, oscillations in control equipment, and ambient laboratory conditions (Proctor et al., 2020). In more extreme cases, cosmic rays originating from outer space have been shown to cause catastrophic multi-qubit errors approximately every ten seconds (McEwen et al., 2022).

Having outlined conformal prediction and quantum machine learning, we now turn to their intersection. We introduce a model for PQC-based learning that incorporates time-dependent noise effects. We then formalise how this temporal variation disrupts the statistical assumptions, particularly the exchangeability of scores, that underlie standard conformal methods, motivating the need for a new approach.

### 3.1 PQCs as Non-stationary Probabilistic Models

To establish the formal setting, let $\mathcal{X}$ and $\mathcal{Y}$ denote a classical feature space and classical target space respectively. For a given feature $x \in \mathcal{X}$, in the angle encoding setting adopted here, a $Q$-qubit PQC prepares

the quantum state

$$|\psi(x)\rangle = U_{N_G}(x) \circ \cdots \circ U_1(x) |0\rangle^{\otimes Q}.$$

Each $U_i(x)$ is a unitary operator parametrised by $x$ representing the ideal $i^{\text{th}}$ gate, and $N_G$ is the number of gates (Nielsen & Chuang, 2010). The specific transformation $U(x) = U_{N_G}(x) \circ \cdots \circ U_1(x)$ depends on the circuit ansatz, any circuit parameters $\theta$, and the data-encoding scheme chosen (see Section 2.2). We suppress these dependencies in the notation for clarity. Define the superoperator $\mathcal{U}_{x,i}(\cdot) \equiv U_i(x)(\cdot)U_i(x)^\dagger$. In an ideal (noiseless) setting, the resulting state is described by the density matrix

$$\rho(x) = |\psi(x)\rangle \langle\psi(x)| = U(x) |0\rangle^{\otimes Q} \langle 0|^{\otimes Q} U(x)^\dagger = \mathcal{U}_{x,N_G} \circ \cdots \circ \mathcal{U}_{x,1}(|0\rangle^{\otimes Q} \langle 0|^{\otimes Q}).$$

When using quantum hardware, the measured state deviates from this ideal due to various noise processes, such as gate errors, decoherence, and crosstalk. Following the convention in Endo et al. (2021), the noisy quantum gate can be written as $\mathcal{E}_t \circ U$, with $\mathcal{E}_t$ being the time-dependent noise channel and $t$ indexing the effective execution time of the circuit shot. This yields the noisy output state

$$\rho_{\text{noisy}}(x, t) = \mathcal{E}_{t,N_G} \circ \mathcal{U}_{x,N_G} \circ \cdots \circ \mathcal{E}_{t,1} \circ \mathcal{U}_{x,1}(|0\rangle^{\otimes Q} \langle 0|^{\otimes Q}).$$

Although in reality individual gates and readout processes occur at different physical times, a single coarse timestamp per shot is enough to establish that the score distribution can vary with time. A computational-basis measurement is performed on the noisy state $\rho_{\text{noisy}}(x, t)$. Since the circuit acts on $Q$ qubits, the measurement yields one of the $2^Q$ bitstring outcomes. To interpret bitstring outcomes in the target space $\mathcal{Y}$, we define a task-dependent mapping

$$f : \{0, 1\}^Q \to \mathcal{Y}.$$

This mapping distributes bitstrings over a grid in $\mathcal{Y}$, with resolution growing as $Q$ increases. Define the random variable

$$\hat{Y}_{x,t} = f(b), \quad \text{whenever the measurement at time } t \text{ with classical input } x \text{ yields } b \in \{0, 1\}^Q.$$

When dealing with noisy quantum measurements, the most comprehensive approach uses a Positive Operator-Valued Measure (POVM) (Nielsen & Chuang, 2010, Box 2.5). For a system of $Q$ qubits, the POVM consists of $2^Q$ elements, denoted as $\{\Pi_j\}$, with each element corresponding to a $Q$-bit measurement outcome $b_j$. The probability of getting a specific outcome $y$ is given by

$$\mathbb{P}(\hat{Y}_{x,t} = y \mid X = x) = \sum_{\{j: f(b_j)=y\}} \text{Tr}(\rho_{\text{noisy}}\Pi_j),$$

due to Born's rule (Born, 1926).

In a perfect, noise-free scenario, each POVM element $\Pi_j$ is simply the projector $|b_j\rangle \langle b_j|$. However, with possible non-stationary noise in the measurement stage, the $\Pi_j$'s can be any positive semidefinite, time-dependent operators that sum to the identity matrix (Bravyi et al., 2021).

A single execution plus measurement (a shot) at time $t$ involves both the noise from the PQC execution and the measurement. Denote the shot from the distribution $\hat{Y}_{x,t}$ by $\hat{y}$. Collecting $M$ such shots, at times $T = \{t_1, \ldots, t_M\}$, produces the sample multiset $\mathcal{A}_{x,T} = \{\!\!\{\hat{y}_m\}\!\!\}_{m=1}^M$. We denote the samples as a multiset to represent potential repeated measurements of the same bitstring. Each shot carries its own timestamp $t_m$, so taking an additional shot, even on the same input, may draw from a different distribution, reflecting the drift in both the execution and measurement noise.

## 3.2 Consequences of Non-stationary Noise on Split Conformal Prediction

In standard settings, a score function is defined as a mapping

$$\hat{S} : \mathcal{X} \times \mathcal{Y} \to \mathbb{R},$$

which assigns a real-valued score to each feature-target pair $(x, y)$. This is typically used to measure the discrepancy between the model output and an observed value. In our case, however, the situation differs: we

obtain a stochastic multiset, $\mathcal{A}_{x,T}$, instead of a single deterministic value. For the score to be a well-defined deterministic function, it is therefore necessary to take $\mathcal{A}_{x,T}$ as an additional input:

$$\hat{S}(x, y\,; \mathcal{A}_{x,T}), \quad \text{with } x \in \mathcal{X},\ y \in \mathcal{Y}.$$

A crucial point is that $\mathcal{A}_{x,T}$ is drawn from a distribution that is conditional on the shot times $T$. As a result, the induced scores are inherently time-dependent if the noise of quantum hardware changes across time. This breaks the usual exchangeability assumption, as even if the underlying data $(X_i, Y_i)$ are exchangeable, the augmented observations

$$Z_i = \big(X_i, Y_i; \{\!\{\hat{Y}_{X_i,t}\}\!\}_{t \in T_i}\big),$$

are not, and therefore the corresponding scores

$$S_i = \hat{S}\big(X_i, Y_i; \{\!\{\hat{Y}_{X_i,t}\}\!\}_{t \in T_i}\big),$$

are not necessarily exchangeable. While a hypothetical score function could be constructed to remove the time dependency (e.g., $\hat{S}(x, y; \{\!\{\hat{Y}_{x,t}\}\!\}_{t \in T}) = \tilde{S}(x, y)$), the conformity score of any QCP procedure is designed to utilise the quantum model's output, and hence should utilise these time-dependent terms.

Without exchangeable scores, we cannot assert that the rank of the test score is uniformly distributed on the set $\{1, \ldots, n+1\}$. Consequently, without making the assumption of stationary noise, we cannot obtain guarantees in the form given in Theorem 1.

## 4   Adaptive Quantum Conformal Prediction

The above section motivates the need for new quantum conformal procedures that are robust to non-exchangeable scores. A substantial body of conformal prediction literature already addresses non-exchangeable feature-target data, which commonly arises in time-series settings. These works provide a natural foundation for adaptation.

Existing approaches can be characterised by the restrictions they impose on the type of shift, varying from known covariate shifts (Tibshirani et al., 2019; Gibbs et al., 2025) to unknown joint distribution shifts (Foygel Barber et al., 2023; Gibbs & Candès, 2021; Xu & Xie, 2021), and by the finite-sample or asymptotic nature of their guarantees.

This section introduces the Adaptive Quantum Conformal Prediction (AQCP) algorithm, the application of the Adaptive Conformal Inference (ACI) framework (Gibbs & Candès, 2021) to the quantum setting. We choose to modify ACI because it makes no assumptions about the form of the underlying distribution shift. The adjustment of ACI to the quantum setting requires the use of quantum-specific score functions and modified assumptions to obtain a finite-sample theoretical result (see Theorem 2).

An alternative approach, discussed in Appendix A, is to examine the conformal guarantees available to quantum models under the framework of Foygel Barber et al. (2023). This framework is particularly relevant as it provides finite-sample guarantees while accommodating arbitrary distribution shift. However, it is not the primary focus here, since obtaining a tight bound within this framework would require quantifying the total variation distance caused by the distribution shift.

### 4.1   AQCP Algorithm

AQCP operates in an online testing setting where the miscoverage level is dynamically adjusted according to observed empirical coverage. As a result, it assumes that the response associated with each test point is revealed before the subsequent test point is processed.

Starting with an initial calibration set $\mathcal{D}_{\text{cal}}$ of size $n$ (see Section 2.1), the algorithm processes test points sequentially. At the $i^{\text{th}}$ test point, AQCP: constructs a prediction set $C_i(X_{n+i}) \subseteq \mathcal{Y}$ using the current miscoverage level $\alpha_i$, observes whether the true label $Y_{n+i}$ falls within this set, updates the miscoverage level to $\alpha_{i+1}$ based on the outcome, and appends the newly observed point $(X_{n+i}, Y_{n+i})$ to the calibration set to

inform future prediction sets. This feedback mechanism allows the algorithm to adapt to shifts in the score distribution without making assumptions about the nature of the shift.

Given a desired miscoverage rate $\alpha \in [0, 1]$, the update to the miscoverage level is

$$\alpha_1 = \alpha,$$
$$\alpha_{i+1} = \alpha_i + \gamma (\alpha - \text{err}_i), \quad i \in \mathbb{N}.$$

Here, $\gamma > 0$ is a step size hyperparameter and the error function, $\text{err}_i$, is given by

$$\text{err}_i := \begin{cases} 1 & \text{if } Y_{n+i} \notin C_i(X_{n+i}), \\ 0 & \text{otherwise.} \end{cases}$$

Each value $\alpha_i$ is used to construct the prediction set $C_i(X_{n+i})$ for $Y_{n+i}$. This set contains all outcomes $y$ whose score does not exceed the $(1 - \alpha_i)$-quantile of the scores from all previous observations. Formally, the prediction set is defined

$$C_i(X_{n+i}) := \left\{ y \in \mathcal{Y} : \hat{S}\left(X_{n+i}, y; \mathcal{A}_{X_{n+i}, T_{n+i}}\right) \leq \text{Quantile}\left(1 - \alpha_i, \frac{1}{n+i} \sum_{j=1}^{n+i-1} \delta_{\hat{S}(X_j, Y_j; \mathcal{A}_{X_j, T_j})} + \delta_{+\infty}\right) \right\}.$$

See Algorithm 1 for the full statement of this process. The choice of step size is important: if it is too high, the coverage becomes too volatile; if it is too low, the system will not adapt fast enough to the changes in the distribution. Choosing $\gamma = 0$ recovers QCP with an updating calibration set.

Notice that the adaptive parameter $\alpha_i$ is proven to remain within a bounded interval $[-\gamma, 1 + \gamma]$ with probability one, ensuring the stability of the method. This stability property is then used to establish a bound on the average miscoverage error over $N$ test points,

$$\left| \frac{1}{N} \sum_{i=1}^{N} \text{err}_i - \alpha \right| \leq \frac{\max\{\alpha_1, 1 - \alpha_1\} + \gamma}{N\gamma}.$$

This bounds the difference between the average observed miscoverage and the target miscoverage $\alpha$. In particular, since the bound decreases inversely with $N$, it follows that as the number of observations increases, the average miscoverage rate converges almost surely to the desired target $\alpha$,

$$\lim_{N \to \infty} \frac{1}{N} \sum_{j=1}^{N} \text{err}_j \overset{\text{a.s.}}{=} \alpha.$$

These guarantees do not depend on exchangeability. This means they are applicable with no assumptions imposed on the noise induced by the quantum hardware. The trade-off for this greater generality is that the guarantees are asymptotic in the number of test points. Gibbs & Candès (2021, Theorem 4.1) give a finite-sample guarantee under a set of specific conditions, namely that the distributional shift is fully determined by a hidden Markov model. This is not directly applicable to our setting, as there is no distributional shift in the feature-target data itself. Instead, the shift arises from the time dependence of the shots which parametrise the score function. However, with a simple adaptation to the assumptions and proof (see Appendix C.1 for the latter), we can state the following theorem.

**Theorem 2 (Finite-Sample Guarantee for AQCP, Adapted from Gibbs & Candès (2021))**
*Let $(X_i, Y_i)$ be i.i.d., and suppose prediction sets are constructed using the empirical $(1 - \alpha_i)$-quantile of conformity scores computed from a fixed calibration dataset $\mathcal{D}_{\mathrm{cal}}$ (Algorithm 1 without the optional step). Assume the test conformity scores are conditionally independent given a hidden Markov chain $(B_i)$ with state space $\mathcal{B}$. Suppose that the joint process $(\alpha_k, B_{n+k})$ forms a Markov chain on $[-\gamma, 1 + \gamma] \times \mathcal{B}$ with unique stationary distribution $\pi$, and that the process is initialised at stationarity. Let $(B_i)$ have transition operator $P_B$ and stationary distribution $\pi_B$ (the marginal of $\pi$ on $\mathcal{B}$). Assume that $P_B$ has non-zero absolute spectral gap[1] $1 - \eta > 0$. Define*

$$B := \sup_{b \in \mathcal{B}} \left| \mathbb{E}[\mathrm{err}_i \mid B_{n+i} = b] - \alpha \right|, \qquad \sigma_B^2 := \mathbb{E}\left[ \left( \mathbb{E}[\mathrm{err}_i \mid B_{n+i}] - \alpha \right)^2 \right].$$

*Then for any $\varepsilon > 0$,*

$$\mathbb{P}\left( \left| \frac{1}{N} \sum_{i=1}^{N} \mathrm{err}_i - \alpha \right| \geq \varepsilon \right) \leq 2 \exp\left( -\frac{N\varepsilon^2}{8} \right) + 2 \exp\left( -\frac{N(1 - \eta)\varepsilon^2}{8(1 + \eta)\sigma_B^2 + 20B\varepsilon} \right).$$

Although the assumptions required by this theorem are unlikely to hold exactly in our setting, we expect the bound to be broadly representative of the algorithm's empirical behaviour, as argued in Gibbs & Candès (2021).

## 4.2 AQCP Score Functions

A core design choice in conformal prediction is that of the score function. Different score functions lead to prediction sets with varying sizes and geometries. Our PQC-based model produces samples from an unknown distribution conditioned upon the model input and time. Therefore, our score functions must use a sample-based approach.

Park & Simeone (2023) achieved excellent results using a k-nearest neighbour (k-NN) score function. For more points of comparison, we turn to the field of probabilistic conformal prediction (Wang et al., 2023; Sadinle et al., 2019; Romano et al., 2020; Angelopoulos et al., 2025). This field defines optimal conditions for a score function, which we discuss in Appendix B. Motivated by that discussion, we consider the following score functions:

- **Euclidean distance:** Let $\bar{y}$ be the mean of the samples $\mathcal{A}_{x,T}$. Define the score function $\hat{S}_{\mathrm{Euc}}(x, y) := \|y - \bar{y}\|_2$.

- **k-nearest neighbour:** Let $\hat{y}_{(k)}$ be the $k^{\mathrm{th}}$ nearest neighbour of $y$ among the samples. Define the score function $\hat{S}_{\mathrm{k\text{-}NN}}(x, y) := \|y - \hat{y}_{(k)}\|_2$.

- **Kernel density estimation:** Using a kernel function $K$ and bandwidth $h$, one first forms a density estimate

$$\hat{p}(y \mid x, \mathcal{A}_{x,T}) = \frac{1}{Mh^d} \sum_{m=1}^{M} K\left( \frac{y - \hat{y}_m}{h} \right).$$

  Define the score function $\hat{S}_{\mathrm{KDE}}(x, y) := -\hat{p}(y \mid x, \mathcal{A}_{x,T})$.

- **High-density region:** Similarly, using the density estimate $\hat{p}(y \mid x, \mathcal{A}_{x,T})$ defined above, the HDR score function is the estimated probability mass of the region where the density is at least as large as the density at the candidate point $y$

$$\hat{S}_{\mathrm{HDR}}(x, y) := \int_{\{y' : \hat{p}(y' \mid x, \mathcal{A}_{x,T}) \geq \hat{p}(y \mid x, \mathcal{A}_{x,T})\}} \hat{p}(y' \mid x, \mathcal{A}_{x,T}) \mathrm{d}y'.$$

---

[1]See Definition 2 for the definition of the absolute spectral gap.

---

**Algorithm 1:** Adaptive Quantum Conformal Prediction (AQCP)

---

**Input** : Miscoverage level $\alpha \in [0,1]$
Score function $\hat{S}$
Initial calibration dataset $\mathcal{D}_{\text{cal}} = \{(x_i, y_i)\}_{i=1}^n$
Test stream $\mathcal{D}_{\text{test}} = \{(x_i, y_i)\}_{i=n+1}^{n+n'}$
Number of quantum shots $M \geq 1$
Step size $\gamma > 0$

**Output:** A sequence of prediction sets for the test stream $\{C_i(x_{n+i})\}_{i=1}^{n'}$

---

**Core Functions** ——————————————————————————————————

**Procedure** *InitialCalibrate($\mathcal{D}_{cal}$)*:
   **for** $(x_i, y_i) \in \mathcal{D}_{cal}$ **do**
      $\mathcal{A}_{x_i, T_i} \leftarrow$ M shots from PQC with input $x_i$ ;
      Add $\hat{S}(x_i, y_i; \mathcal{A}_{x_i, T_i})$ to $\mathcal{S}$ ;

**Function** *GetQuantile($\mathcal{S}, \alpha$)*:
   **return** $\inf \left\{ q : \frac{1}{|\mathcal{S}|} \sum_{s_i \in \mathcal{S}} \mathbb{1}\{s_i \leq q\} \geq 1 - \alpha \right\}$ ;

**Function** *GeneratePredictionSet($x, \lambda, \mathcal{A}_{x,T}$)*:
   Initialise prediction set $C(x) \leftarrow \emptyset$ ;
   **foreach** $y \in \mathcal{Y}$ **do**
      **if** $\hat{S}(x, y; \mathcal{A}_{x,T}) \leq \lambda$ **then**
         Add $y$ to $C(x)$ ;
   **return** $C(x)$ ;

**Procedure** *UpdateState($x, y, \lambda, \mathcal{A}_{x,T}$)*:
   $s \leftarrow \hat{S}(x, y; \mathcal{A}_{x,T})$ ;
   **if** $s > \lambda$ **then**
      err $\leftarrow 1$ ;
   **else**
      err $\leftarrow 0$ ;
   $\alpha \leftarrow \alpha + \gamma(\alpha_1 - \text{err})$ ;
   Add $s$ to $\mathcal{S}$ ;                                    ▷ % Optional Step %

---

**Main Algorithm Execution** ————————————————————————————

PredictionSets $\leftarrow \emptyset$ ;
$\mathcal{S} \leftarrow \emptyset$ ;                                                 ▷ % Score set %
$\alpha_1 \leftarrow \alpha$ ;
InitialCalibrate($\mathcal{D}_{\text{cal}}$) ;
**for** $i = 1$ **to** $n'$ **do**
   $\lambda \leftarrow$ GetQuantile($\mathcal{S}, \alpha$) ;
   $\mathcal{A}_{x_{n+i}, T_{n+i}} \leftarrow$ M shots from PQC with input $x_{n+i}$ ;
   $C_i(x_{n+i}) \leftarrow$ GeneratePredictionSet($x_{n+i}, \lambda, \mathcal{A}_{x_{n+i}, T_{n+i}}$) ;
   Add $C_i(x_{n+i})$ to PredictionSets ;
   UpdateState($x_{n+i}, y_{n+i}, \lambda, \mathcal{A}_{x_{n+i}, T_{n+i}}$) ;
**return** PredictionSets ;

---

## 5 Experiments

The following experiments implement Adaptive Quantum Conformal Prediction (AQCP) (Algorithm 1) on a univariate multimodal regression task. We test local coverage properties of AQCP using data from the `ibm_sherbrooke` quantum processor. Additionally, we investigate the impact of the score function and the number of shots on prediction set size using data from a classical simulator.

Code and data for reproducing our experiments is available at: `https://github.com/doug-spencer/AQCP`. All `ibm_sherbrooke` shot data were collected on the 18th of April 2025.

### 5.1 Experimental Setup

To facilitate comparison with Park & Simeone (2023), we replicate the conditions of their regression task, in which training, calibration, and test data are drawn i.i.d. from the following distribution:

$$X \sim \mathcal{U}(-10, 10), \qquad Y \mid X = x \sim \frac{1}{2}\Big(\mathcal{N}(-\mu(x), 0.05^2) + \mathcal{N}(\mu(x), 0.05^2)\Big),$$

where $\mathcal{U}(-10, 10)$ denotes the uniform distribution on $[-10, 10]$, and $\mathcal{N}(m, \sigma^2)$ denotes the univariate normal distribution with mean $m$ and variance $\sigma^2$. The function $\mu(x)$ is defined as

$$\mu(x) = \frac{1}{2}\sin\left(\frac{4}{5}x\right) + \frac{x}{20}.$$

#### 5.1.1 Quantum Model Architecture and Training

To apply AQCP, a trained model must first be obtained. In the classical-data quantum-processing paradigm we follow, this requires specifying a parametrised quantum circuit (PQC), a classical data-encoding scheme, and an optimisation procedure. Background on this is provided in Section 2.2 and our approach parallels that of Park & Simeone (2023).

The hardware-efficient ansatz (HEA) was selected for its versatility, problem-agnostic design, and hardware efficiency (Simeone, 2022). We implemented the HEA using $Q = 5$ qubits and $L = 5$ layers applied sequentially to form the unitary operator

$$U(\theta) \coloneqq U_5(\theta) \cdot U_4(\theta) \cdot U_3(\theta) \cdot U_2(\theta) \cdot U_1(\theta).$$

Each layer was formed of an unparametrised entangling unitary, $U_{\text{ent}}$, and parametrised single-qubit Pauli rotation gates:

$$U_l(\boldsymbol{\theta}) \coloneqq U_{\text{ent}}\big(R_Z(\theta_{l,1}^1)R_Y(\theta_{l,1}^2)R_Z(\theta_{l,1}^3) \otimes \cdots \otimes R_Z(\theta_{l,Q}^1)R_Y(\theta_{l,Q}^2)R_Z(\theta_{l,Q}^3)\big), \quad l = 1, \ldots, 5,$$

$$U_{\text{ent}} \coloneqq \prod_{k=1}^{Q-1} C_{k,k+1}^Z.$$

Here, $C_{k,k+1}^Z$ denotes a controlled-$Z$ gate between qubits $k$ and $k+1$, forming a linear entangling block (see Section 2.2.1). Classical features were encoded using a learned non-linear angle encoding scheme with data re-uploading. Specifically, a neural network with architecture $(1, 10, 10, |\boldsymbol{\theta}|)$, where $|\boldsymbol{\theta}| = 3LQ = 75$, maps each input $x$ to the circuit rotation angles $\boldsymbol{\theta_W}(x)$. The network includes bias terms and uses Exponential Linear Unit (ELU) activations (Clevert et al., 2015); $\boldsymbol{W}$ denotes its trainable weights.

All PQC measurements were performed in the computational basis. Bitstring outcomes $b \in \{0, 1\}^Q$ were then mapped to a discrete real-valued grid via $f : \{0, 1\}^Q \to \mathbb{R}$, defined

$$f(b) = y_{\min} + k \cdot \text{bin}(b), \quad \text{with} \quad k = \frac{y_{\max} - y_{\min}}{2^Q - 1}.$$

Here, $\text{bin}(b)$ converts bitstrings to their denary representation, and $[y_{\min}, y_{\max}] = [-1.5, 1.5]$ was chosen to contain all but a negligible fraction of the probability mass of the target distribution.

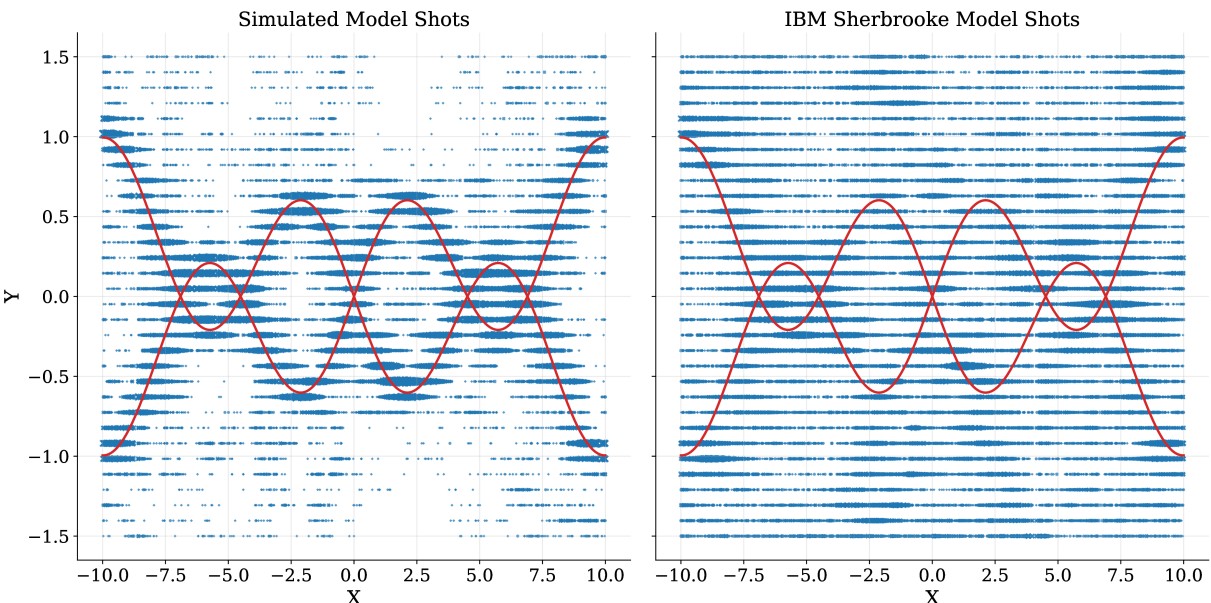

Figure 2: **Regression model shots (simulated vs. `ibm_sherbrooke`).** Comparison of 100,000 shots sampled from each backend (Qiskit Aer simulator and `ibm_sherbrooke`). The marker size is scaled proportionally to the count of overlapping shots at each location. The red lines represent the component mean functions $\mu(x)$ and $-\mu(x)$.

We trained the angle encoder parameters using the TorchQuantum framework (Wang et al., 2022), which integrates the construction and simulation of quantum circuits with PyTorch's automatic gradient computation. While this method of training would not be possible using quantum hardware, alternative methods such as the parameter-shift rule (Schuld et al., 2019; Park & Simeone, 2023) are available.

TorchQuantum also provides direct access to the full measurement distribution of the PQC as a probability mass function over bitstrings, enabling the use of a multi-class cross-entropy loss. For a single training example $(x_i, y_i)$, this is defined as

$$\ell\left(y_i, U(\boldsymbol{\theta_W}(x_i))\right) := -\log\left(\mathbb{P}(\hat{Y} = y_i \mid x_i)\right),$$

where $\hat{Y}$ denotes the PQC measurement outcome. All training was performed on a noiseless simulator, so we do not condition on shot time in this instance.

The model was trained to minimise the empirical risk,

$$\hat{R}(\boldsymbol{W}; \mathcal{D}_{\mathrm{tr}}) := \frac{1}{n_{\mathrm{tr}}} \sum_{i=1}^{n_{\mathrm{tr}}} \ell\left(y_i, U\left(\boldsymbol{\theta_W}(x_i)\right)\right),$$

with $\mathcal{D}_{\mathrm{tr}} = \{(x_i, y_i)\}_{i=1}^{n_{\mathrm{tr}}}$, $n_{\mathrm{tr}} = 1,000$, a fixed learning rate of 0.01, and 100 epochs.

Figure 2 shows a scatter plot of points taken using the trained model on a noiseless simulator and using the `ibm_sherbrooke` backend. It is clear from the simulated samples that the model reasonably approximates the conditional distribution. While the fundamental structure of the model remains discernible, the samples from `ibm_sherbrooke` show the clear impact of hardware noise.

### 5.1.2 Algorithm Implementation and Evaluation Strategy

To generate calibration and test data, we drew 10,000 samples from the target distribution. For each sample, we executed the circuit with the encoded parameters on `ibm_sherbrooke` for $M = 100$ shots. These 10,000

circuits were submitted to the device in batches of 1,000. In the efficiency studies, we collected shot data for 10,000 samples, but from the classical simulator `FakeQuitoV2`. All data were collected through the Qiskit library (Javadi-Abhari et al., 2024).

When implementing AQCP, we used Algorithm 1 (with the optional update step included), meaning that conformity scores from each test point were appended to the calibration set after evaluation. All score functions were implemented as introduced in Section 4.2. $k = \lceil \sqrt{M} \rceil$ was chosen for the $\hat{S}_{\text{k-NN}}$ score function, where $M$ is the shot number. This is in line with the choice in Park & Simeone (2023), and with the asymptotic bounds required for consistency. For the $\hat{S}_{\text{KDE}}$ and $\hat{S}_{\text{HDR}}$ score functions, the Gaussian kernel was implemented and the bandwidth was chosen via Silverman's 'rule of thumb'. When calculating scores, additional Gaussian noise with $\sigma = 10^{-4}$ was added to break ties. Without this tie-breaking step, a large number of scores would have been equal due to the discrete nature of the grid mapping to output points.

Coverage properties are evaluated using a rolling window of recent test points, which preserves sensitivity to transient undercoverage that may be obscured by long-run averages. To evaluate the effect of adaptivity, we compare AQCP with its zero-step-size counterpart ($\gamma = 0$). This corresponds to applying QCP with an expanding calibration dataset and no feedback adjustment, which we refer to as *online QCP*.

In the efficiency study, we consider only AQCP with a positive step size. To provide a notion of optimality, we define $\mathcal{C}^*$ as the class of prediction sets that minimise the expected prediction set size subject to the marginal coverage constraint. Further details on this optimality criterion are given in Appendix B.1.

## 5.2 Local Coverage Results

The efficacy of the adaptive recalibration mechanism in AQCP is now demonstrated. Figures 3 and 4 illustrate the local coverage over a stream of test points for the score functions introduced with a moving average. The blue lines represent the online QCP algorithm (equivalent to AQCP with an adaptation step size of $\gamma = 0$). The orange lines represent the AQCP algorithm with $\gamma = 0.03$.

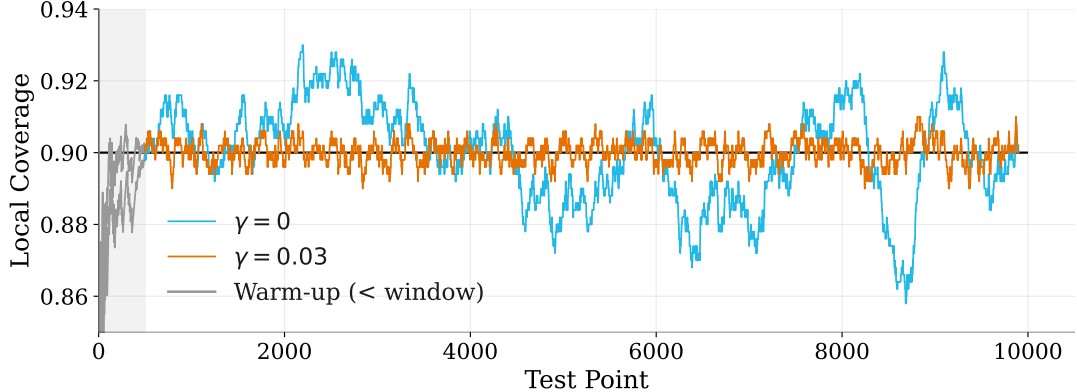

Figure 3: **Moving average coverage of AQCP ($\gamma = 0, 0.03$) on the multimodal regression task using shot data from `ibm_sherbrooke`.** The k-NN score function. 100 initial calibration points are used with a rolling window of size 500, and a target miscoverage of $\alpha = 0.1$.

Figure 3 presents the results obtained using the k-NN score function. The baseline of online QCP exhibits substantial deviations from the target coverage of $\alpha$. For example, it over-covers between the 2,000–3,000 test points, and later under-covers around test point 8,500. In contrast, AQCP shows greater stability. Once the initial rolling window is fully populated, AQCP consistently maintains the average coverage around the desired 90% target level. The algorithm's ability to dynamically adjust its miscoverage estimate allows it to counteract prolonged over- or under-coverage.

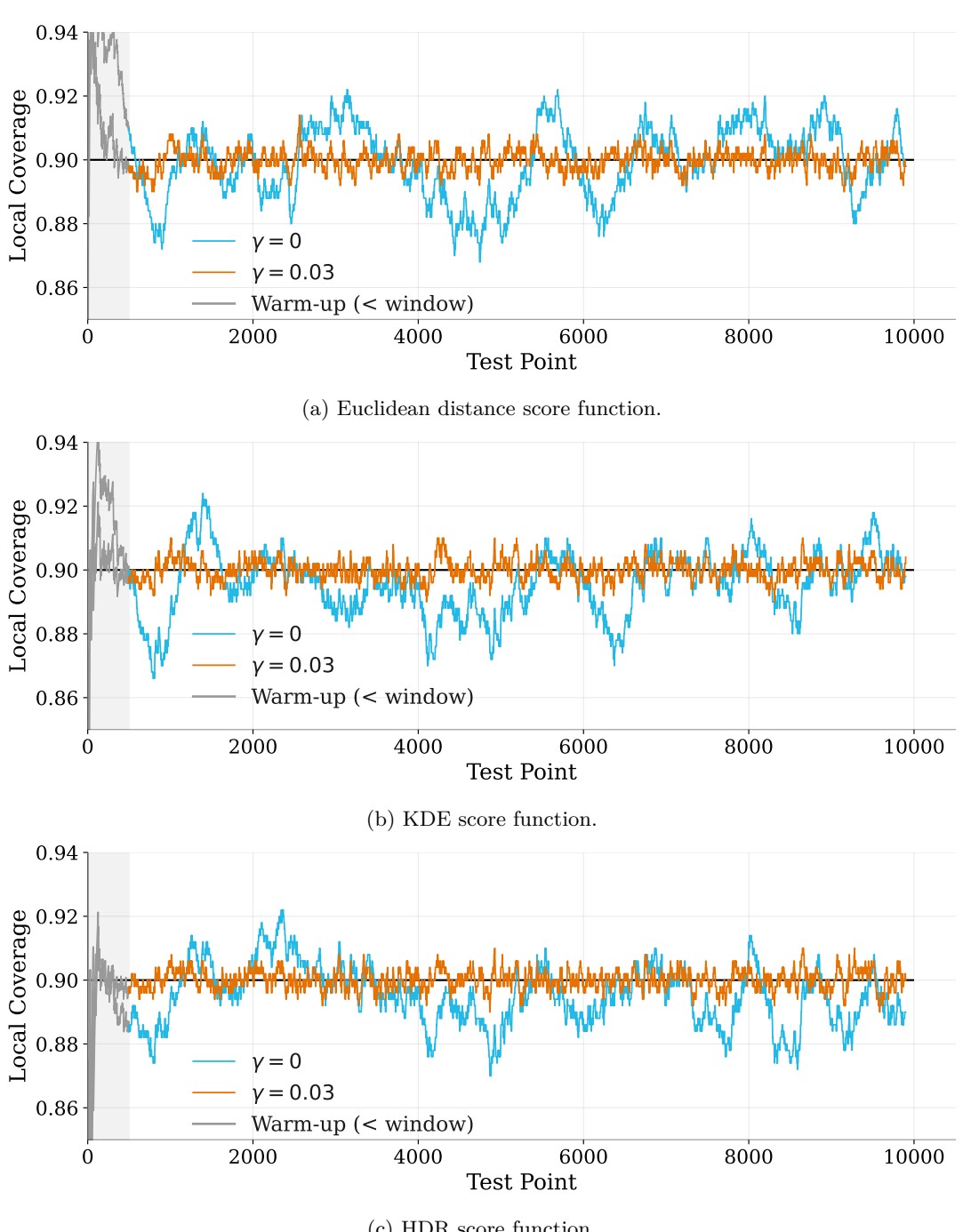

(a) Euclidean distance score function.

(b) KDE score function.

(c) HDR score function.

Figure 4: **Moving average coverage of AQCP ($\gamma = 0$, $0.03$) on the multimodal regression task using shot data from `ibm_sherbrooke`.** Euclidean distance, KDE, and HDR score functions. 100 initial calibration points are used with a rolling window of size 500, and a target miscoverage of $\alpha = 0.1$.

Similarly, Figure 4 demonstrates the robust stabilising behaviour of AQCP across the Euclidean distance ($\hat{S}_{\text{Euc}}$), kernel density estimation ($\hat{S}_{\text{KDE}}$), and high-density region ($\hat{S}_{\text{HDR}}$) score functions. In all cases, the AQCP algorithm ($\gamma = 0.03$) consistently maintains coverage closer to the nominal level than the online QCP algorithm ($\gamma = 0$). This demonstrates that the choice of the score function does not substantially influence the coverage stability achieved by AQCP.

### 5.3 Efficiency Results

We now focus on the efficiency of AQCP set predictors, specifically examining the impact of the score function and the number of shots $M$ on the average prediction set size. Since non-stationary noise is not directly relevant to this analysis, all shots were generated using the `FakeQuitoV2` backend. To isolate the effect of the score function and shot number, the step size is fixed at $\gamma = 0.03$. Results are presented as piecewise linear curves of the average set size across ten logarithmically spaced shot values ranging from $M = 1$ to $M = 1{,}000$.

Figure 5(a) presents the efficiency results for our multimodal regression task. All score functions perform similarly for small shot numbers $M \leq 10$, after which their behaviours diverge. $\hat{S}_{\text{KDE}}$ and $\hat{S}_{\text{HDR}}$ produce comparable average set sizes across all values of $M$, both showing a steady decline in average set size as $M$ increases logarithmically. $\hat{S}_{\text{HDR}}$ achieves the smallest average set size at $M = 1{,}000$. $\hat{S}_{\text{KDE}}$ demonstrates a more rapid decrease in the medium $M$ range but plateaus for $M \geq 100$. $\hat{S}_{\text{Euc}}$ and $\hat{S}_{\text{1-NN}}$ exhibit different behaviour. While comparable performance was observed at small shot numbers ($M \leq 10$), they both produced substantially larger prediction sets across the entire range of $M$ values. At $M = 1{,}000$, $\hat{S}_{\text{Euc}}$ yielded prediction sets approximately 1.5–2 times larger than those produced by $\hat{S}_{\text{HDR}}$, despite both achieving the target coverage level of 90%, as shown in Figure 5(b).

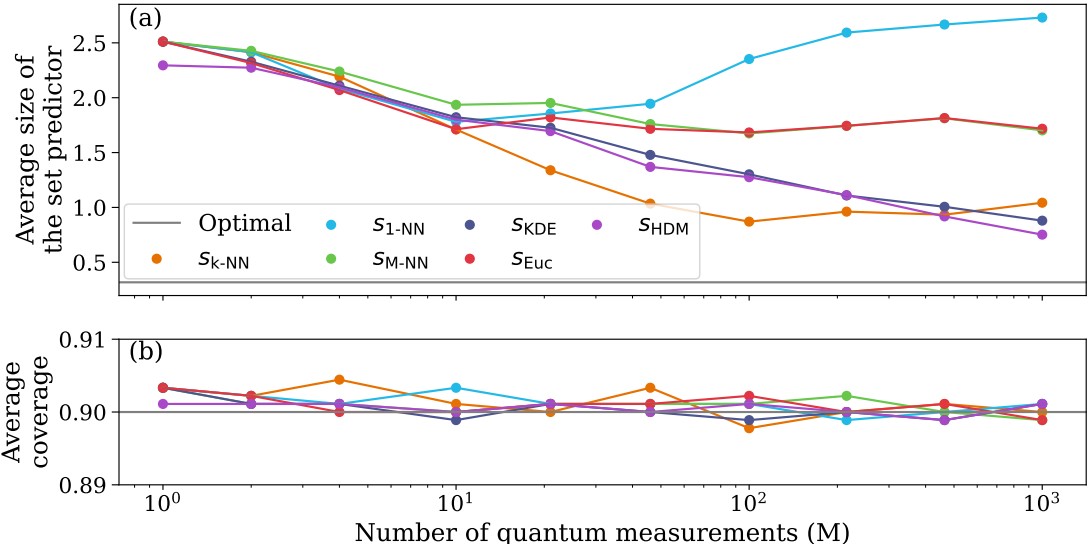

Figure 5: **Average coverage and average set size of AQCP ($\gamma = 0.03$), evaluated across a range of shot numbers $M$ using shot data from `FakeQuitoV2`.** The desired miscoverage is set to $\alpha = 0.1$. Averages are computed from prediction sets returned from Algorithm 1 with 100 initial calibration points and 9,900 test points. (a) Average prediction set size for a range of score functions. (b) Corresponding average coverage for the same score functions. The optimal line represents the performance of the benchmark $\mathcal{C}^*$ family of prediction sets.

### 5.4 Limitations and Future Work

The empirical evaluation of robustness to non-stationary noise is based on hardware runs collected over a single day and therefore does not fully characterise longer-term drift or more structured perturbations. While the observed behaviour is consistent with the theoretical motivation for adaptive calibration, a more comprehensive assessment across multiple days, together with evaluation under controlled noise perturbations, would provide a stronger empirical stress-test.

The behaviour of AQCP would be further clarified by a systematic step-size sensitivity analysis, as well as the incorporation of step-size schedules proposed for Adaptive Conformal Inference in related literature (Podkopaev et al., 2024; Zaffran et al., 2022; Bhatnagar et al., 2023) in place of the fixed step size used here. We view such extended evaluation as an important direction for future work. Furthermore, empirical evaluation of the method of Foygel Barber et al. (2023) (described in Appendix A) may provide additional insight into robustness beyond the exchangeable setting.

Finally, we note that AQCP is inherently sequential: it updates the miscoverage level using observed outcomes and thus requires that each test response be revealed before the next prediction is made. It is therefore not directly applicable in batch prediction settings. Developing methods that are valid under non-stationary quantum noise in batch settings remains an open direction for future work.

## 6 Conclusion

This work addresses a critical challenge for the reliability of quantum conformal prediction: the non-stationary nature of hardware noise in current-generation quantum processors. The analysis formalised why, without the assumption of stationary noise, the standard conformal guarantee may not hold.

We drew from the existing literature to propose alternative theories that provide conformal guarantees without assuming stationary noise. Among these, Gibbs & Candès (2021) emerged as the most applicable candidate, and their algorithm was implemented using shot data from a trained QML model. The conditions of the multimodal regression experiment in Park & Simeone (2023) were reproduced, and coverage was observed to oscillate closely around the target level for both the QCP framework (with an updating calibration dataset) and the AQCP algorithm. Notably, AQCP exhibited greater stability around the target coverage level. Confidently attributing the greater deviation of QCP from the target coverage level to non-stationary noise would require further experimentation, since fluctuations are expected when testing with finite samples. Nevertheless, these findings support the use of AQCP for quantum models when an online calibration procedure is feasible.

AQCP was also implemented with a variety of sample-based score functions, including those examined by Park & Simeone (2023). The performance of these score functions aligns with the findings in Park & Simeone (2023), and both newly introduced score functions (kernel density estimation-based and high-density region-based) obtained near-optimal set sizes with a high number of shots. This outcome is also consistent with the theory in Appendix B.

### Acknowledgements

D.S. is grateful to Professor Osvaldo Simeone of Northeastern University London for his initial guidance that helped frame the scope of this work and for providing feedback on an early draft. S.N. also extends thanks to Dr. Paul Dellar of Corpus Christi College, University of Oxford, for his mentorship and guidance.

Additionally, the authors acknowledge the use of IBM Quantum services for this work. The views expressed are those of the authors and do not reflect the official policy or position of IBM or the IBM Quantum team.

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

# A   Appendix: Conformal Prediction Beyond Exchangeability

We follow the procedure in Foygel Barber et al. (2023), making the necessary adaptations to our setting. Denote $Z_i = (X_i, Y_i; \mathcal{A}_{X_i, T_i})$. A weight $w_i \in [0, 1]$ is assigned to each data point to quantify its similarity to a given test point. These weights can be derived from various metrics, such as the temporal gap between the observation and the test instance. The underlying intuition is to assign higher weights to data points presumed to share the same distribution as the test point, $(X_{n+1}, Y_{n+1}; \mathcal{A}_{X_{n+1}, T_{n+1}})$, and lower weights to those from different distributions. The weights are subsequently normalised,

$$\tilde{w}_i = \frac{w_i}{\sum_{j=1}^{n} w_j + 1}, \quad \text{for} \quad i = 1, \dots, n \qquad \tilde{w}_{n+1} = \frac{1}{\sum_{j=1}^{n} w_j + 1}.$$

Foygel Barber et al. (2023) provide equivalent theoretical bounds for both non-symmetric full conformal and split-conformal algorithms. We concentrate only on the split-conformal case. The non-exchangeable split conformal set is given by

$$C(X_{n+1}) = \left\{ y \in \mathcal{Y} : \hat{S}(X_{n+1}, y; \mathcal{A}_{X_{n+1}, T_{n+1}}) \leq \text{Quantile}\left(1 - \alpha, \sum_{i=1}^{n} \tilde{w}_i \cdot \delta_{\hat{S}(Z_i)} + \tilde{w}_{n+1} \cdot \delta_{+\infty}\right) \right\}.$$

With this prediction set, we can obtain the finite-sample guarantee from Foygel Barber et al. (2023)

$$\mathbb{P}(Y_{n+1} \in C(X_{n+1})) \geq 1 - \alpha - \sum_{i=1}^{n} \tilde{w}_i \cdot \mathrm{d}_{\mathrm{TV}}(S(Z), S(Z^i)),$$

where $Z = (Z_1, \dots, Z_{n+1})$ represents the original data sequence, $Z^i = (Z_1, \dots, Z_{i-1}, Z_{n+1}, Z_{i+1}, \dots, Z_n, Z_i)$ represents the same sequence but with the $i^{\text{th}}$ and $(n + 1)^{\text{th}}$ observations swapped, $S(z) \in \mathbb{R}^{n+1}$ is the residual vector with entries $(S(z))_i = \hat{S}(x_i, y_i; \mathcal{A}_{x_i, T_i})$, and $\mathrm{d}_{\mathrm{TV}}$ denotes the total variation distance. Their method protects against shifts in the distributions of the $Z_i$'s and, as a consequence, shifts in the scores.

The method provides a coverage guarantee of at least $1 - \alpha$ minus a specific correction term. This correction reflects how much the data deviate from exchangeability, weighted by the importance given to each observation. If the data are truly exchangeable, this correction becomes zero, restoring the standard $1 - \alpha$ guarantee. However, in QML, the size of this correction is in practice unknown. This limits the ability to adjust the weights to counteract large distributional shifts, meaning no practical lower bound on coverage can be given without making further assumptions.

## B   Appendix: Score Functions

The effectiveness of any conformal method, specifically the size of the resulting prediction sets, is highly dependent on the score function. This section therefore delves into the theory of optimal score functions to construct prediction sets that are as small as possible while maintaining coverage. We then introduce practical, sample-based score estimators suitable for the probabilistic output of a quantum model.

### B.1   What Makes a Score Function Optimal?

The conformal prediction literature considers several optimality criteria. We follow Angelopoulos et al. (2025), focusing on two natural objectives: constructing prediction sets with minimal expected size subject to either marginal or conditional coverage.

Let $(\Omega, \mathcal{F}, \mathbb{P})$ be a probability space, and let $X$ and $Y$ be random variables taking values in Borel subsets $\mathcal{X} \subseteq \mathbb{R}^{d_X}$ and $\mathcal{Y} \subseteq \mathbb{R}^{d_Y}$, equipped with their respective Borel $\sigma$-algebras. Let $P_{X,Y}$ denote their joint law and $P_X$ the marginal law of $X$. Consider measurable joint prediction sets $B \in \mathcal{F}_{\mathcal{X}} \otimes \mathcal{F}_{\mathcal{Y}}$, with sections

$$B(x) \coloneqq \{y : (x, y) \in B\}.$$

Let $|B(x)|$ denote the Lebesgue measure on $\mathcal{Y}$. Given a miscoverage level $\alpha \in (0, 1)$, the two optimisation problems are:

1. **Marginal coverage objective:**

$$\underset{B}{\operatorname{argmin}} \, \mathbb{E}\left[|B(X)|\right] \quad \text{subject to} \quad \mathbb{P}\big(Y \in B(X)\big) \geq 1 - \alpha.$$

2. **Conditional coverage objective:**

$$\underset{B}{\operatorname{argmin}} \, \mathbb{E}\left[|B(X)|\right] \quad \text{subject to} \quad \mathbb{P}\big(Y \in B(x) \mid X = x\big) \geq 1 - \alpha \quad \text{for } P_X\text{-a.e. } x.$$

Assume that $P_{X,Y}$ is absolutely continuous with respect to Lebesgue product measure on $\mathcal{X} \times \mathcal{Y}$, with joint density $p(x, y)$, marginal density $p_X(x) > 0$ Lebesgue-a.e., and conditional density

$$p(y \mid x) = \frac{p(x, y)}{p_X(x)}.$$

For simplicity, assume that the relevant density values have no flat spots: for the marginal problem, $p(Y \mid X)$ has no atoms under $P_{X,Y}$; for the conditional problem, for $P_X$-a.e. $x$, the random variable $p(Y \mid x)$ under $Y \mid X = x$ has no atoms.

In conformal prediction, prediction sets are typically constructed as sublevel sets of a score function $\hat{S}(x, y)$:

$$C_\lambda(x) \coloneqq \{y \in \mathcal{Y} : \hat{S}(x, y) \leq \lambda\},$$

where $\lambda \in \mathbb{R}$ is calibrated to achieve the desired coverage. For any fixed prediction set $B$, choosing

$$\hat{S}(x, y) = 1 - \mathbb{1}_{B(x)}(y), \qquad \lambda = 0,$$

gives $C_\lambda(x) = B(x)$. The optimal-score problem is therefore the stronger problem of identifying scores whose sublevel sets generate optimal prediction sets across all coverage levels.

**Definition 1 (Optimal score functions)** *A score function $\hat{S} : \mathcal{X} \times \mathcal{Y} \to \mathbb{R}$ is optimal with respect to optimisation problem i if, for every $\alpha \in (0,1)$, there exists $\lambda \in \mathbb{R}$ such that $C_\lambda$ is an optimal solution to problem i, up to null modifications. The class of such score functions is denoted by $\mathcal{S}_i$.*

We now give a simple sufficient condition for optimality under the marginal coverage objective.

## B.2 Optimal Scores for Marginal Coverage ($\mathcal{S}_1$)

Adapting arguments from Lei (2014); Sadinle et al. (2019); Kato et al. (2023), we show that any strictly decreasing transformation of the conditional density yields an optimal score for the marginal coverage objective.

**Theorem 3 (Sufficient condition for marginal optimality)** *Under the assumptions above, suppose there exists a strictly decreasing function $\phi : [0, \infty) \to \mathbb{R}$, such that*

$$\hat{S}(X,Y) = \phi(p(Y \mid X))$$

*$P_{X,Y}$-almost surely. Then $\hat{S} \in \mathcal{S}_1$.*

For the proof see Appendix C.2. A similar argument is also presented in Sadinle et al. (2019), but for the case of discrete $\mathcal{Y}$.

Theorem 3 states that any score function that is a strictly decreasing transformation of the conditional probability density is optimal for producing minimal prediction sets with marginal coverage. By exploring various choices for $\phi$, score functions that satisfy this sufficient condition can be recovered, ranging from well-known forms to more nuanced variants. For instance, applying $\phi(x) = -x$ yields $\hat{S} = -p(y \mid x)$; applying $\phi(x) = x^{-1}$ on $x > 0$ (with appropriate handling at $x = 0$) yields $\hat{S} = p(y \mid x)^{-1}$; and choosing $\phi(x) = -\log(x)$ on $x > 0$ (again with appropriate handling at $x = 0$) produces the negative log density score.

In many machine learning settings, any of these forms can be implemented directly. In our setting, the true conditional density $p(y \mid x)$ is unknown and hence it must be estimated from PQC shots.

In classical regression over $\mathbb{R}^n$, there is a particularly appealing connection to the widely used Euclidean distance score function,

$$\hat{S}_{\text{Euc}}(x,y) = \|y - f(x)\|_2,$$

where $f(x)$ is a point prediction model. Theorem 3 implies that $\hat{S}_{\text{Euc}}$ is marginally optimal whenever there exists a strictly decreasing function $\phi$ such that

$$\phi\big(p(y \mid x)\big) = \|y - f(x)\|_2$$

for $P_{X,Y}$-almost every $(x,y)$. Equivalently, on the relevant support,

$$p(y \mid x) = \phi^{-1}\Big(\|y - f(x)\|_2\Big).$$

Thus, the commonly used score function $\|y - f(x)\|_2$ is optimal for marginal coverage in the sense of optimisation problem 1 whenever the conditional density is radially symmetric about $f(x)$, with a radial profile that is decreasing as the distance from $f(x)$ increases and does not depend on $x$. This condition is satisfied, for example, when $f(x)$ is the conditional mean of a homoscedastic isotropic Gaussian model, but it generally fails in the presence of skewed conditional distributions, anisotropy, or heteroscedasticity.

To deal with more general distributions, we look towards probability density estimators. For example, given an appropriate choice of $k$, $\hat{S}_{\text{k-NN}}$ can be viewed as a proxy for a negative k-NN density estimator (Zhao & Lai, 2022; Loftsgaarden & Quesenberry, 1965). Under suitable consistency conditions, this suggests that $\hat{S}_{\text{k-NN}}$ asymptotically approaches a score in the sufficient optimality class as the number of samples $M \to \infty$. Similarly, since kernel density estimation is a consistent density estimator under an appropriate choice of bandwidth $h$ (Parzen, 1962; Devroye & Penrod, 1986; Davis et al., 2011), $\hat{S}_{\text{KDE}}$ can also asymptotically approach this class. This theory assumes access to samples taken from the true conditional distributions; however, it provides motivation for the general case.

## B.3 Optimal Scores for Conditional Coverage ($\mathcal{S}_2$)

For the conditional guarantee optimisation problem, a similar approach can be taken. We first recall the high-density level set (Hyndman, 1996):

$$H_x(t) = \{y \in \mathcal{Y} : p(y \mid x) \geq t\}.$$

This is the set of all outcomes $y$ that are at least as probable as the threshold $t$. Using this gives rise to the next theorem, which provides a sufficient condition for a score function to attain the conditional guarantee.

**Theorem 4 (Sufficient condition for conditional optimality)** *Suppose there exists a strictly increasing function $\phi : [0,1] \to \mathbb{R}$ such that*

$$\hat{S}(x,y) = \phi\left(\int_{H_x(p(y|x))} p(y' \mid x)\mathrm{d}y'\right)$$

*for $P_{X,Y}$-almost every $(x,y) \in \mathcal{X} \times \mathcal{Y}$. Then $\hat{S} \in \mathcal{S}_2$.*

For the proof see Appendix C.3. A similar argument is also presented in Angelopoulos et al. (2025); Romano et al. (2020) but for the case of discrete $\mathcal{Y}$.

Compared with the simpler sufficient condition for $\mathcal{S}_1$, this form is more abstract and can be harder to reduce to immediately implementable scores. However, in parametric models the level-set probability often has a closed form. For example, in regression over $\mathbb{R}$, suppose that for each $x \in \mathcal{X}$ we have

$$Y \mid X = x \sim \mathcal{N}(\mu(x), \sigma^2(x)),$$

where $\mathcal{N}(\mu(x), \sigma^2(x))$ denotes the univariate normal distribution with mean $\mu(x)$ and variance $\sigma^2(x)$. Then the level set corresponding to density value $p(y \mid x)$ is the symmetric interval around $\mu(x)$ with radius $|y - \mu(x)|$. Hence,

$$\mathbb{P}\big(Y \in H_x(p(y \mid x)) \mid X = x\big) = 2\Phi\left(\frac{|y - \mu(x)|}{\sigma(x)}\right) - 1,$$

where $\Phi$ denotes the standard normal CDF. Therefore any score of the form

$$\hat{S}(x,y) = \phi\left(2\Phi\left(\frac{|y - \mu(x)|}{\sigma(x)}\right) - 1\right),$$

for some strictly increasing $\phi : [0,1] \to \mathbb{R}$, satisfies the sufficient condition above. In particular, choosing

$$\phi(z) = \Phi^{-1}\left(\frac{z+1}{2}\right),$$

yields the score

$$\hat{S}(x,y) = \frac{|y - \mu(x)|}{\sigma(x)},$$

which parallels the result from Section B.2. Moreover, this construction extends to any symmetric distribution with a strictly monotonic probability density function in the radial distance from the mean.

When the form of the conditional distribution is unknown, we again revert to a sample-based probability density estimator. Using this estimator and taking $\phi(x) = x$, we then obtain the $\hat{S}_{\mathrm{HDR}}$ score. Under suitable consistency conditions, this score asymptotically approaches a score satisfying the sufficient condition for $\mathcal{S}_2$ as $M \to \infty$.

## B.4 Ties to Adaptive Quantum Conformal Prediction

The optimisation problems and results above do not depend on a conformal construction. Rather, they characterise optimal prediction sets through level sets of score functions, and identify *subclasses* of score

functions that attain optimality under the marginal and conditional coverage objectives when the true conditional density $p(y \mid x)$ is known. Therefore this connection serves as structural guidance in the quantum conformal setting, as opposed to a formal guarantee. The analysis does not quantify how approximation error in $p(y \mid x)$, finite measurement effects, or calibration variability influence the resulting set sizes.

However, under ideal conditions — where measurements closely reflect $Y \mid X$ and the calibration dataset is sufficiently large to estimate thresholds accurately — applying QCP with the described score functions is expected to produce prediction sets that are close to optimal in expectation. The same reasoning extends to AQCP. Since the optimality statement holds for any fixed miscoverage level $\alpha \in (0,1)$, they apply equally to the time-varying levels $\alpha_i$ specified by the adaptive procedure.

## C  Proofs

**Definition 2** *(Absolute Spectral Gap (Jiang et al., 2018)) A $\pi$-invariant Markov operator $P$ has non-zero absolute spectral gap $1 - \lambda(P)$ if*

$$\lambda(P) = \sup \left\{ \|Ph\|_\pi : \|h\|_\pi = 1, \ h \in \mathcal{L}_2^0 \right\} < 1.$$

**Lemma 1** *(Neyman–Pearson) Let $f$ and $g$ be non-negative measurable functions, with $g > 0$ almost everywhere. Suppose there exists $t \geq 0$ such that*

$$\int_{\{f/g \geq t\}} f = 1 - \alpha.$$

*Then $B_t = \{f/g \geq t\}$ is an optimiser of the problem*

$$\min_B \int_B g \quad subject \ to \quad \int_B f \geq 1 - \alpha.$$

### C.1  Proof of Theorem 2

The argument follows Appendix A.7 of Gibbs & Candès (2021), with the environment chain $A_t$ replaced by the score chain $B_i$. We can write

$$\mathbb{P}\left( \left| \frac{1}{N} \sum_{i=1}^N \mathrm{err}_i - \alpha \right| > \varepsilon \right) \leq \mathbb{P}\left( \left| \frac{1}{N} \sum_{i=1}^N (\mathrm{err}_i - \mathbb{E}[\mathrm{err}_i \mid B_{n+i}]) \right| > \frac{\varepsilon}{2} \right)$$
$$+ \mathbb{P}\left( \left| \frac{1}{N} \sum_{i=1}^N (\mathbb{E}[\mathrm{err}_i \mid B_{n+i}] - \alpha) \right| > \frac{\varepsilon}{2} \right).$$

We will first bound the first term using the Hoeffding bound. The proof of Lemma A.2 in the original text uses only the binary-valued errors and the monotonicity of $\alpha_i$ in past errors, $\sum_{s=1}^{i-1} \mathrm{err}_s$. This argument is unchanged here, yielding

$$\mathbb{P}\left( \left| \frac{1}{N} \sum_{i=1}^N (\mathrm{err}_i - \mathbb{E}[\mathrm{err}_i \mid B_{n+i}]) \right| > \frac{\varepsilon}{2} \right) \leq 2 \exp\left( -N\varepsilon^2/8 \right).$$

Now we focus on bounding the second term. Define $f(b) = \mathbb{E}[\mathrm{err}_i \mid B_{n+i} = b] - \alpha$. Then $f$ is bounded in $[-B, B] \subseteq [-1, 1]$, mean-zero under stationarity, and the variance proxy is $\sigma_B^2$. Applying the Bernstein inequality for Markov chains (Theorem A.1 in the original text) yields

$$\mathbb{P}\left( \left| \frac{1}{N} \sum_{i=1}^N (\mathbb{E}[\mathrm{err}_i \mid B_{n+i}] - \alpha) \right| > \frac{\varepsilon}{2} \right) \leq 2 \exp\left( -\frac{N(1-\eta)\varepsilon^2}{8(1+\eta)\sigma_B^2 + 20B\varepsilon} \right).$$

Combining the two bounds proves the theorem.

## C.2 Proof of Theorem 3

For $t \geq 0$, define
$$H(t) := \{(x, y) \in \mathcal{X} \times \mathcal{Y} : p(y \mid x) \geq t\},$$

and
$$h(t) := \mathbb{P}\big((X, Y) \in H(t)\big) = \mathbb{P}\big(p(Y \mid X) \geq t\big).$$

Then $h$ is non-increasing, with $h(0) = 1$ and $\lim_{t \to \infty} h(t) = 0$.

By the no-flat-spots assumption, $h$ is continuous. Hence, for every $\alpha \in (0, 1)$, there exists a threshold $t_\alpha$ such that $h(t_\alpha) = 1 - \alpha$.

Set $B_\alpha := H(t_\alpha)$, and let $f(x, y) := p(x, y)$, and $g(x, y) := p_X(x)$. Then
$$\frac{f(x, y)}{g(x, y)} = p(y \mid x),$$

so
$$B_\alpha = \left\{ (x, y) \in \mathcal{X} \times \mathcal{Y} : \frac{f(x, y)}{g(x, y)} \geq t_\alpha \right\}.$$

Moreover,
$$\int_{B_\alpha} g(x, y) \, dx \, dy = \int_{\mathcal{X}} p_X(x) |B_\alpha(x)| \, dx = \mathbb{E}_X[|B_\alpha(X)|],$$

and
$$\int_{B_\alpha} f(x, y) \, dx \, dy = \mathbb{P}\big((X, Y) \in B_\alpha\big) = \mathbb{P}\big(Y \in B_\alpha(X)\big) = h(t_\alpha) = 1 - \alpha.$$

Therefore, by Lemma 1, $B_\alpha$ is an optimiser of the marginal coverage problem.

It remains to show that $B_\alpha$ is a sublevel set of $\hat{S}$. By assumption, there exists a strictly decreasing function $\phi : [0, \infty) \to \mathbb{R}$ such that
$$\hat{S}(X, Y) = \phi(p(Y \mid X))$$

$P_{X,Y}$-almost surely. Since $\phi$ is strictly decreasing,
$$p(y \mid x) \geq t_\alpha \iff \phi(p(y \mid x)) \leq \phi(t_\alpha).$$

Thus, setting $\lambda_\alpha := \phi(t_\alpha)$, we have
$$B_\alpha = \{(x, y) : \hat{S}(x, y) \leq \lambda_\alpha\}$$

up to $P_{X,Y}$-null sets. Hence, for every $\alpha \in (0, 1)$, there exists a threshold $\lambda_\alpha$ such that the conformal sublevel set $C_{\lambda_\alpha}$ solves the marginal coverage objective. Therefore $\hat{S} \in \mathcal{S}_1$.

## C.3 Proof of Theorem 4

For each $x \in \mathcal{X}$, define the conditional high-density region
$$H_x(t) := \{y : p(y \mid x) \geq t\},$$

and let
$$h_x(t) := \int_{H_x(t)} p(y' \mid x) \mathrm{d}y'.$$

Note that $h_x$ is non-increasing, with $h_x(0) = 1$ and $\lim_{t \to \infty} h_x(t) = 0$.

By the no flat spots assumption, $h_x(t)$ is continuous for $P_X$-almost every $x$. Hence, for every $\alpha \in (0, 1)$, there exists $t_{\alpha,x}$ such that
$$h_x(t_{\alpha,x}) = 1 - \alpha$$

for $P_X$-almost every $x$.

Define the prediction set

$$B_\alpha(x) := H_x(t_{\alpha,x}).$$

By construction,

$$\mathbb{P}\big(Y \in B_\alpha(x) \mid X = x\big) = h_x(t_{\alpha,x}) = 1 - \alpha$$

for $P_X$-almost every $x$. Moreover, as a conditional high-density region, $B_\alpha(x)$ has the minimal Lebesgue measure among all measurable sets with conditional probability at least $1 - \alpha$. Therefore $B_\alpha = \{(x,y) : y \in B_\alpha(x)\}$ is a solution to optimisation problem 2 at level $\alpha$.

It remains to show that $B_\alpha$ is a sublevel set of $\hat{S}$. By assumption, there exists a strictly increasing function $\phi : [0,1] \to \mathbb{R}$ such that

$$\hat{S}(x,y) = \phi\big(h_x(p(y \mid x))\big)$$

for $P_{X,Y}$-almost every $(x,y)$.

Now fix $(x,y) \in \mathcal{X} \times \mathcal{Y}$. We have

$$
\begin{aligned}
y \in B_\alpha(x) &\iff p(y \mid x) \geq t_{\alpha,x} \\
&\iff H_x\big(p(y \mid x)\big) \subseteq H_x(t_{\alpha,x}) \\
&\iff h_x(p(y \mid x)) \leq h_x(t_{\alpha,x}) = 1 - \alpha,
\end{aligned}
$$

where the second equivalence follows from the monotonicity of level sets and the last equivalence holds for $P_{X,Y}$-almost every $(x,y)$.

Since $\phi$ is strictly increasing,

$$h_x(p(y \mid x)) \leq 1 - \alpha \iff \phi\big(h_x(p(y \mid x))\big) \leq \phi(1 - \alpha).$$

Thus, setting

$$\lambda_\alpha := \phi(1 - \alpha),$$

we have

$$B_\alpha(x) = \{y : \hat{S}(x,y) \leq \lambda_\alpha\}$$

for $P_X$-almost every $x$, up to conditional null sets. Hence, for every $\alpha \in (0,1)$, there exists a threshold $\lambda_\alpha$ such that the conformal sublevel set $C_{\lambda_\alpha}$ solves the conditional coverage objective. Therefore $\hat{S} \in \mathcal{S}_2$.

