# OpenReview forum: "Adaptive Conformal Prediction for Quantum Machine Learning"
_TMLR — Accepted by TMLR_

### Review · Reviewer_rrQD · 2026-01-08

**Summary Of Contributions:**

This paper studies uncertainty quantification in quantum machine learning that has provable guarantees in the presence of non-stationary quantum hardware noise. The work is very nicely written in the sense that it introduces all the essential ingredients in a self-contained manner.

**Audience:**

Yes

**Audience Explanation:**

Quantum machine learning has gaining a growing attention in machine learning community for its great potential to provide innovations that classical computing cannot bring us. However, lack of reliability due to non-trivial quantum hardware noise prevents its practical deployment in the real world. This work nicely addresses such limitation by adequately applying modern statistical frameworks in the presence of non-stationary hardware noise.

**Broader Impact Concerns:**

Not applicable.

**Claims And Evidence:**

Yes

**Claims Explanation:**

All the theoretical claims in this work seem to be very solid and rigorous. Also, it makes a lot of sense that quantum hardware noise will suffer from non-stationarity, hence breaking the exchangeability, hence making the classical split conformal prediction no longer valid in general. However, the current experimental design used in Fig. 3-4 seems somewhat inappropriate to support such a claim.

In Fig. 3-4, the authors used the metric of local coverage -- time-averaged coverage with increased, augmented, calibration set -- to compare split conformal (SC) and adaptive conformal inference (ACI). However, such metric is in favor of ACI, not SC. Let us hereby recall that SC guarantees marginal coverage with "random" calibration data set with "fixed" size. However, local coverage in Fig. 3-4 assume "deterministic" calibration data set (given that the same calibration data points are keep used and never being refreshed) with "increasing" size. I believe that, a fairer comparison with SC and ACI would be to compare the usual marginal coverage -- e.g., start with bunch of data points, run quantum machine learning for each data point to get multiple shots, save such data set, and consider random calibration-test split to get multiple realizations of coverage, average over such multiple coverage values to report the estimated marginal coverage. If the quantum shots across different data points suffer from sufficient "non"-exchangeability, then such (usual) experimental design would be enough to show the supremacy of the proposed method.

In short, I think Fig. 3-4 is too in favor of the proposed method given that it considers time-averaged, deterministic, coverage. It would be beneficial to have some additional plot that is in favor of split conformal, e.g., marginal coverage, to further support the claim of this work.

Minor comments

- The results in the appendix seems super interesting especially regarding the part using Neyman-Pearson lemma to prove that conditional distribution would provide a scoring function that is optimal in the sense of the prediction set size. I also notice that the authors mentioned that all the chosen scoring functions are all consistent density estimator hence asymptotically optimal in the  sense of prediction set size, which is great. But it seems that the results in the appendix (that uses Neyman-Pearson lemma) assumes split conformal prediction (as it assumes usual marginal coverage) and it is not clear how this is related to the ACI setting considered in this work. That said, would optimal scoring for split conformal still be optimal for ACI?

- In Theorem 2, what is "quantile estimator"? Also, what is absolute spectral gap? Lastly, I think we should state the stationary condition (\alpha_1, B_1)\sim \pi before defining B and \sigma_B^2.

- I recommend checking some of the nice papers by Maria Schuld such as [Schuld M, Bergholm V, Gogolin C, Izaac J, Killoran N. Evaluating analytic gradients on quantum hardware. Physical Review A. 2019 Mar;99(3):032331.] that firstly proposed parameter-shift rule (to the best of my knowledge); and [Schuld M. Supervised quantum machine learning models are kernel methods. arXiv preprint arXiv:2101.11020. 2021 Jan 26.] for the "CQ" paradigm of quantum machine learning, to enrich the citations made in this work.

**Requested Changes:**

As mentioned above, please consider applying the following changes:

- experimental design that shows marginal coverage, not time-averaged coverage
- can the optimality of scoring functions proven for split conformal be applied for ACI case?

---

> ### Author Response · Authors · 2026-02-25
> **Response to reviewer rrQD**
>
> We sincerely thank the reviewer for the in-depth feedback.
>
> __On marginal coverage with random calibration-test splits__
>
> The reviewer suggested evaluating usual marginal coverage by repeatedly drawing random calibration-test splits from a fixed set of datapoints and quantum-shots. We considered this approach during the early stages of this work, and agree it marks a more usual test setting for split conformal prediction. However, we believe that in our setting such a procedure would introduce an artificial exchangeability that masks the phenomenon under study.
>
> Concretely, suppose we collect a dataset $((X_i, Y_i, shots_i))_{i=1}^n$ where the shot data are generated sequentially under time-varying hardware noise. Performing random calibration-test splits on this fixed dataset amounts to selecting a subset of indices uniformly at random without replacement to form the calibration set, with the remaining indices forming the test set. Conditioned on the realised dataset, this is equivalent to sampling without replacement from a finite population, which is exchangeable by construction: every ordering of the selected elements is equally likely.
>
> An alternative approach that would avoid this would be to re-run the quantum hardware to obtain fresh shots for each calibration-test split. However, hardware constraints severely limit the number of such repetitions that we could have performed, so we opted against this route.
>
> __On comparison to split conformal methods__
>
> We acknowledge the reviewer’s point that split conformal prediction (SC) is typically analysed under marginal coverage with a fixed-size random calibration set, whereas our experiments evaluate sequential local coverage with an expanding calibration set.
>
> Our goal is precisely to evaluate performance under sequentially arriving data where exchangeability may fail, which is the setting when the adaptive mechanism can be applied. We acknowledge that this was somewhat unclear in the original version and so in the revised version we have both clarified this (opening paragraph of Section 4.1 in the revised manuscript) and included an Experimental Limitations and Future Work section (Section 5.4 in the revised manuscript) that details the limitations of AQCP to settings where errors are known after each time step.
>
> When data arrive sequentially, SC would naturally be applied online using an updating calibration dataset. In this setting, we believe it is appropriate to compare QCP with an online expanding calibration dataset to AQCP. This reflects the deployment scenario we envision with the goal being maximising stability around the target level.
>
> __Appendix Results__
>
> We thank the reviewer for highlighting this point. We have clarified the connection between the optimality results and AQCP in Appendix B.4 of the revised manuscript. To summarise this, the Neyman-Pearson analysis is not tied to split conformal prediction itself, but instead characterises prediction sets through score-function level sets under ideal knowledge of the conditional density $p(y \mid x)$. The optimality result therefore concerns the geometry of level sets for any prescribed miscoverage level, independently of how that level is selected. In this sense, the structural optimality results provide guidance for both QCP and AQCP: adaptivity governs how $\alpha_n$ evolves, but optimal scoring governs the efficiency of the resulting sets at each step.
>
> __Theorem 2__
>
>  We thank the reviewer for their detailed questions, which helped us improve the precision of Theorem 2. First, we clarified the terminology around the “quantile estimator,'' explicitly defining it as the empirical $(1-\alpha_i)$-quantile of conformity scores computed from a fixed calibration dataset with a connection to Algorithm 1. Second, we added a formal definition for the absolute spectral gap (Definition 1 in appendix C which we obtained from the Jiang et al., 2018 - cited in the revised version). Finally, we reordered the theorem statement as suggested, ensuring the stationary condition and initialisation are introduced prior to defining $B$ and $\sigma_B^2$.
>
> __Additional references__
>
> We appreciate the reviewer pointing out these foundational works, and we agree they enrich our citations. We have incorporated both papers into the revised manuscript. Schuld et al. (2019) has been added to Section 5.1.1 where we suggest alternative methods for training the model, such as the parameter-shift rule. Schuld (2021) is now cited twice: once in Section 2.2 regarding the "CQ" paradigm, and again in Section 2.2.2 in our explicit discussion of angle encoding.

---

### Review · Reviewer_hfYK · 2026-01-19

**Summary Of Contributions:**

The paper studies the problem of uncertainty quantification in a quantum setting. The existing methods can ensure the exchangability property (i.e., invariance of the joint under permuting) of the conformity scores only under simplifying assumptions. The authors adapt a classical algorithm to remove this assumption. An implementation of the algorithm is then evaluated on IBM's quantum hardware with varying score functions, attaining better stability than the state of the art.

**Audience:**

Yes

**Audience Explanation:**

The studied questions appear very fundamental even if outside my area of expertise, so I'm sure there'd be many interested individuals for this paper.

**Broader Impact Concerns:**

A Broader Impact Statement is missing but also not needed in my opinion.

**Claims And Evidence:**

Yes

**Claims Explanation:**

The paper is very detailed, but due to it being outside my area of expertise, I wasn't able to assess the correctness of all theoretical details nor the meaningfulness of the experiments. The earlier parts which I could verify seemed correct, and the figures show the improved stability convincingly.

**Requested Changes:**

Minor:
- I think it would be better to define exchangeability already in Sec. 2.1 (or earlier) since understanding Theorem 1 requires it.
- I'd be more explicit about that $S_i$'s are random variables and not some functions; I got mildly confused by the differences between $s_i$, $S_i$, and $\hat{S}$ at some point.
- Start of p. 4: unnecessary colon
- Start of p. 6: citation not in parentheses

---

> ### Author Response · Authors · 2026-02-24
> **Response to reviewer hfYK**
>
> We sincerely thank the reviewer for their response. We have addressed the requested minor changes in the revised version as follows:
>
> __Exchangeability definition:__ While the full formal definition remains in Section 2.2, we have added a concise summary in Section 2.1 at the point where exchangeability is first introduced, so that Theorem 1 can be understood without forward reference.
>
> __Clarification of notation:__ We have added an explicit clarification in Section 2.1 that uppercase letters $X_i, Y_i, S_i$ denote random variables and lowercase letters denote their realised values. We also now define the score function explicitly using $\hat{S}:\mathcal{X}\times\mathcal{Y}\to\mathbb{R}$,  to make the distinction between the score function and scores clearer.
>
> __Minor stylistic corrections:__ We have removed the unnecessary colon at the start of p. 4 and corrected the citation formatting at the start of p. 6. and appreciate these suggestions from the reviewer.

---

### Review · Reviewer_4WJG · 2026-02-11

**Summary Of Contributions:**

The paper shows that non-stationary quantum hardware noise breaks the exchangeability needed for standard conformal guarantees, then adapts Adaptive Conformal Inference to define Adaptive Quantum Conformal Prediction (AQCP). It evaluates AQCP on IBM hardware and in simulation, compares score functions, the findings align with the related work.

**Audience:**

Yes

**Audience Explanation:**

AQCP is a practical wrapper for existing quantum models and relevant to the quantum machine learning community.

**Claims And Evidence:**

Yes

**Claims Explanation:**

The theory adapts results from Gibbs and Candes. Experiments replicate Park and Simeone’s multimodal regression setup and use real IBM hardware data (ibm_sherbrooke, April 2025), where AQCP’s coverage oscillates near target with improved stability. Evidence for robustness to non-stationary noise is limited by single-day data; multi-day or perturbed-noise tests and step-size sensitivity would strengthen the claim.
In addition to numeric evidence, the authors make their code available.
Open code is an important addition to the paper. Systematic testing of key functions using `pytest` ( https://docs.pytest.org/en/stable/ ) could have strengthened the paper further.

**Requested Changes:**

None

---

> ### Author Response · Authors · 2026-02-24
> **Response to reviewer 4WJG**
>
> We sincerely thank the reviewer for their feedback.
>
> >Evidence for robustness to non-stationary noise is limited by single-day data; multi-day or perturbed-noise tests and step-size sensitivity would strengthen the claim.
>
> In the revised manuscript, we have added a dedicated Experimental Limitations and Future Work section (5.4 of revised version) that explicitly acknowledges these limitations and highlights relevant literature to guide future analyses of step-size sensitivity for AQCP.
>
> We also thank the reviewer for the suggestion regarding systematic testing using tools such as pytest. We agree that an automated test suite would further strengthen the reproducibility and robustness of the codebase. We will take this into account in future work and subsequent development of AQCP.

---

### Decision · Action_Editor_cCef · 2026-04-07

**Recommendation:** Accept as is

**Additional Comments:**

The authors addressed each of the reviewer concerns during the rebuttal phase. Pertinently, a new limitations and future work section was added to ensure that the claims were properly scoped. After the changes, the reviewers were unanimous in their opinion to accept the paper.

**Audience:**

Yes

**Audience Explanation:**

There is an increasing level of interest in quantum machine learning. The proposed algorithm, which addresses reliability concerns due to quantum hardware noise, is therefore likely to be of interest to the community. AQCP can be applied as a wrapper on top of existing algorithms. Code is provided, which is likely to facilitate adoption.

**Claims And Evidence:**

Yes

**Claims Explanation:**

This paper proposes a new algorithm called Adaptive Quantum Conformal Prediction (AQCP) for uncertainty quantification in the quantum machine learning setting. Previous approaches to this problem do not  take into account the non-stationarity of quantum hardware noise. AQCP builds upon Adaptive Conformal Inference (ACI) in order to provide theoretically-grounded reliability guarantees in the non-stationary quantum setting. The proposed algorithm was evaluated both on IBM hardware and in simulation, showing improved stability.